# The 123 COVID SCORE: A simple and reliable diagnostic tool to predict in-hospital death in COVID-19 patients on hospital admission

Małgorzata Ostrowska[1]*, Michał Kasprzak[1], Tomasz Fabiszak[1], Jacek Gajda[2], Natalia Jaje-Rykowska[3], Piotr Michalski[1], Beata Moczulska[3], Paulina Nowek[3], Maciej Piasecki[1], Marta Pilaczyńska-Cemel[4], Przemysław Podhajski[1], Paulina Prudzic[1], Dominika Stępniak[3], Damian Świątkowski[1], Maciej Żechowicz[3], Robert Gajda[2], Leszek Gromadziński[3], Jacek Kryś[1], Aldona Kubica[1], Grzegorz Przybylski[4], Paweł Szymański[5], Jacek Kubica[1]

**1** Collegium Medicum, Nicolaus Copernicus University, Bydgoszcz, Poland, **2** Gajda-Med District Hospital in Pultusk, Pultusk, Poland, **3** Department of Cardiology and Internal Medicine, School of Medicine, Collegium Medicum, University of Warmia and Mazury, Olsztyn, Poland, **4** Department of Lung Diseases, Neoplasms and Tuberculosis, Faculty of Medicine, Nicolaus Copernicus University, Bydgoszcz, Poland, **5** Department of Cardiology, Interventional Cardiology and Electrophysiology with Cardiac Intensive Care Unit, Tertiary Care Hospital, Grudziądz, Poland

* m.ostrowska@cm.umk.pl

**Data Availability Statement:** Data cannot be shared publicly because of ethical concerns. Data are available from the corresponding author, or

## Abstract

### Background

Patients hospitalized due to Coronavirus disease 2019 (COVID-19) are still burdened with high risk of death. The aim of this study was to create a risk score predicting in-hospital mortality in COVID-19 patients on hospital admission.

### Methods

Independent mortality predictors identified in multivariate logistic regression analysis were used to build the 123 COVID SCORE. Diagnostic performance of the score was evaluated using the area under the receiver-operating characteristic curve (AUROC).

### Results

Data from 673 COVID-19 patients with median age of 70 years were used to build the score. In-hospital death occurred in 124 study participants (18.4%). The final score is composed of 3 variables that were found predictive of mortality in multivariate logistic regression analysis: (1) age, (2) oxygen saturation on hospital admission without oxygen supplementation and (3) percentage of lung involvement in chest computed tomography (CT). Four point ranges have been identified: 0–5, 6–8, 9–11 and 12–17, respectively corresponding to low (1.5%), moderate (13.4%), high (28.4%) and very high (57.3%) risk of in-hospital death. The 123 COVID SCORE accuracy measured with the AUROC was 0.797 (95% CI 0.757–0.838; p<0.0001) in the study population and 0.774 (95% CI 0.728–0.821; p<0.0001) in an external validation cohort consisting of 558 COVID-19 patients.

from the following non-author institutional points of contact, for researchers who meet the criteria for access to confidential data. kikkardiol@cm.umk.pl / tel. 0048525854023 This is an e-mail address and telephone number to Department of Cardiology and Internal Medicine, Collegium Medicum, Nicolaus Copernicus University, Bydgoszcz, Poland. 2.piotr.adamski@cm.umk.pl / tel. 0048525854023 This is an e-mail address and telephone number to non-author Associate professor in Department of Cardiology and Internal Medicine, Collegium Medicum, Nicolaus Copernicus University, Bydgoszcz, Poland. We will ensure persistent and long-term data storage and availability with full data set deposited and available by contact with either corresponding author or two provided non-author contacts. One of them being institutional and durable point of contact, that ensures data will be accessible even if an author changes email addresses, institutions, or becomes unavailable to answer requests.

**Funding:** The author(s) received no specific funding for this work.

**Competing interests:** The authors have declared that no competing interests exist.

## Conclusions

The 123 COVID SCORE containing merely 3 variables: age, oxygen saturation, and percentage of lung involvement assessed with chest CT is a simple and reliable tool to predict in-hospital death in COVID-19 patients upon hospital admission.

## Introduction

Since the outbreak of the coronavirus disease 2019 (COVID-19) pandemic the World Health Organization (WHO) has recorded 773,449,299 confirmed cases and 6,991,842 deaths, with mortality rate of about 5.3% [1]. Due to community-wide transmission of the severe acute respiratory syndrome coronavirus 2 (SARS-CoV2), unpredictable course of the COVID-19, ranging from asymptomatic to life-threatening, and lack of causative treatment the disease has become one of the greatest healthcare challenges of our times [2–8]. According to WHO severity definitions the following forms of COVID-19 can be distinguished: i) critical COVID-19 defined by the criteria for acute respiratory distress syndrome, sepsis, septic shock or any condition requiring life sustaining treatment; ii) severe COVID-19 defined by oxygen saturation <90% on room air, signs of pneumonia or signs of severe respiratory distress; iii) non-severe COVID-19 defined as absence of any criteria for severe or critical COVID-19 [9]. The availability of COVID-19 vaccines helps to prevent symptomatic, severe and critical disease [10]. However, as of December 2023 the WHO still reports each day hundreds of COVID-19-related deaths [1]. Statistical analyses from multiple countries place COVID-19 among three leading causes of death just after cardiovascular diseases and cancer [11–13]. Furthermore, the COVID-19 Excess Mortality Collaborators estimate the real-life mortality rates to be three-fold higher than those reported, especially in the regions of South Asia, North Africa and the Middle East, and eastern Europe [14]. The risk of in-hospital death due to COVID-19 may reach up to 30% [15], therefore identification of patients burdened with the highest risk of death is of paramount importance. The extremely high mortality rates recorded in northern Italy during the first wave of the COVID-19 pandemic were primarily associated with old age, lung disease and heart failure, resulting in this threesome together with COVID-19 to be known as the deadly quartet [16]. According to the European Centre for Disease Prevention and Control the risk of very severe disease increases with advancing age, presence of comorbidities and male sex [17]. The clinical characteristics of COVID-19 patients predictive of death or admission to a critical care unit included older age, male sex, non-caucasian ethnicity, comorbidities together with higher respiratory rate, lower blood oxygenation and severe radiographic changes [18]. Multiple laboratory test were found to predict in-hospital death in COVID-19 patients [19–21]. An Italian report by Ruscica et al. points to N-terminal pro-B-type natriuretic peptide, interleukin 6 and lactate dehydrogenase as predictors of in-hospital death [19]. Yet, Ye et al. found decreased albumin level and elevated concentrations of liver biochemical markers such as aspartate aminotrasferase, alanine aminotrasferase, total bilirubin and lactic dehydrogenase to correlate with COVID-19-related mortality [20]. In a meta-analysis by Mesas et al. including 60 studies with a total of 51,225 patients from hospitals in 13 countries, decreased platelet count, decreased hemoglobin concentration, increased creatinine, increased interleukin 6 and increased cardiac troponin I were found to be associated with a higher risk of in-hospital death in COVID-19 patients ≤60 years of age [21]. On the other hand, in patients >60 years of age decreased albumin, increased total bilirubin, aspartate aminotrasferase, alanine aminotrasferase, urea nitrogen, C-reactive protein, lactate dehydrogenase and ferritin predicted higher mortality.

In our previous publication we identified age ≥70 years, oxygen saturation ($SpO_2$) ≤87% on admission without oxygen supplementation, lungs involvement in computed tomography (CT) ≥40%, and concomitant coronary artery disease (CAD) as independent predictors of in-hospital death in COVID-19 patients [22]. Although multiple different predictors of unfavorable outcomes in COVID-19 patients have been identified so far, we are still looking for a simple, user-friendly and reliable tool to assess the risk of in-hospital death in our patients. Therefore, in the current study we took another step ahead and based on the results from our previous analysis we propose a risk score for prediction of in-hospital death in COVID-19 patients.

## Materials and methods

This is a retrospective analysis of prospectively collected data from a registry of COVID-19 patients hospitalized in University Hospital No. 1 in Bydgoszcz and University Hospital in Olsztyn between February 1st, 2021 and December 31st, 2022. The registry was designed to gather comprehensive information on consecutive patients hospitalized due to SARS-CoV-2 infection confirmed with Real Time Polymerase Chain Reaction (RT-PCR) test. The following data were collected: demographic information, past medical history, clinical presentation on hospital admission, laboratory test results, computed tomography results, course of the COVID-19 infection (including demand for oxygen support, non-invasive ventilation and mechanical ventilation), and treatment. Data collection was performed by treating physicians thus at the time of therapy provision it was impossible to blank patients' identity. However, information collected in the database do not include any data that could directly identify individual participants. Based on the results from our previous publication, factors independently predicting in-hospital death were used to build a 123 COVID SCORE [22]. The study was conducted in accordance with the Declaration of Helsinki and was approved by the Local Ethics Committee (study approval reference number KB 6/2021 and KB 24/2021). It is a retrospective study of medical reports, thus the participants' consents were not obtained. The study did not include minors.

### External validation

External validation of the 123 COVID SCORE was performed in COVID-19 patients (with SARS-CoV-2 infection confirmed in RT-PCR test) hospitalized in three Polish hospitals: in the Kuyavian-Pomeranian Center of Pulmonology in Bydgoszcz, in Regional Specialist Hospital in Grudziądz, and in Gajda-Med Medical Center in Pułtusk between February 1st, 2021 and December 31st, 2022.

### Statistical analysis

Statistical analysis was performed using the Statistica 13.0 package (TIBCO Software Inc, California, USA). The Shapiro-Wilk test demonstrated non-normal distribution of the investigated continuous variables. Therefore, continuous variables were presented as medians with interquartile range and non-parametric Mann-Whitney U test was used. Categorical variables were expressed as numbers and percentages and were compared using the χ2 test. In order to identify predictors for mortality, multivariate logistic regression model was implemented. To create a score for prediction of mortality risk, a scoring module of Statistica package was used. To select variables for building the model, the backward stepwise logistic regression method was used. Then, continuous variables were reclassified using the weight of evidence method. A scoring table was built, based on which the final prediction model was created. The diagnostic performance of the score was assessed by creating receiver operating characteristic (ROC)

curves and measuring the area under the ROC curve (AUROC) and their 95% confidence intervals. Results were considered significant at p<0.05.

## Results

The total study population included in our registry consisted of 1040 consecutive COVID-19 patients hospitalized in University Hospital No. 1 in Bydgoszcz (59.6%) and University Hospital in Olsztyn (40.4%) between October 2020 and May 2021. According to multivariate analysis, independent predictors of in-hospital death included: age ≥70 years, oxygen saturation on hospital admission ≤87% measured without oxygen supplementation, percentage of lung involvement ≥40% as assessed with chest CT, and presence of coronary artery disease. After exclusion of all records with missing data, of the final analysis on which establishment of the 123 COVID SCORE was based included 673 patients. The median age of study participants was 70 years (ranging from 60 to 79 years). A vast majority of patients (88.9%) demanded oxygen supplementation, of whom 15.3% required non-invasive ventilation support and 12.8% mechanical ventilation support. In-hospital death occurred in 124 study participants (18.4%). Baseline characteristics of the study group are presented in Table 1.

Factors that had been identified in our previous multivariate analysis as predictors of in-hospital death were now included in a multivariate logistic regression model. After introducing age, SpO$_2$ on hospital admission without oxygen supplementation and percentage of lung involvement according to chest CT as continuous variables into the mulivariate logistic regression analysis, CAD became statistically insiginificant and was removed from the model (Table 2). Only variables serving as independent predictors of mortality were incorporated into the risk score (Table 3).

The 123 COVID SCORE ranges from 0 to 17 points. The higher the score, the poorer the outcome. After establishing the cut-off values, four risk groups were proposed, i.e. 0–5, 6–8, 9–11 and 12–17 points, respectively corresponding to. low (1.5%), moderate (13.4%), high (28.4%)and very high (57.3%) risk of death (Fig 1).

The in the 123 COVID SCORE is very simple to use and takes only 3 steps as shown in Fig 2.

The diagnostic accuracy of the 123 COVID SCORE as assessed with the AUROC was 0.797 (95% CI 0.757–0.838; p<0.0001) (Fig 3).

Validation cohort was composed of 558 COVID-19 patients (339 males and 219 females) with a median age of 66 years (56–75). In-hospital death occurred in 141 cases (25.3%). The diagnostic accuracy of the 123 COVID SCORE in the validation cohort as assessed with the AUROC was 0.774 (95% CI 0.728–0.821; p<0.0001) (Fig 4).

## Discussion

The reported risk of in-hospital death in patients hospitalized due to COVID-19 may reach up to 30% [15, 23]. Since the very beginning of the COVID-19 pandemic multiple variables were found to predict disease severity and risk of in-hospital death [19–22]. Besides advanced age and male sex, the European Centre for Disease Prevention and Control lists hypertension, diabetes, chronic kidney disease, coronary heart disease, chronic obstructive pulmonary disease, cerebrovascular disease and chronic liver disease, use of immunosuppressive medications, arrythmia, ischemic heart disease, heart failure, cancer, obesity, and smoking as risk factors for severe disease [17]. The United States Centers for Disease Control and Prevention add few other medical conditions increasing the risk of severe COVID-19 including: cystic fibrosis, dementia or other neurological conditions, pregnancy, disability, human immunodeficiency virus infection, mental health conditions, physical inactivity, sickle cell disease or thalassemia,

**Table 1. Baseline clinical characteristics of study group.** Variables are presented as median (interquartile range) or number (%).

| Variable | All patients (n = 693) | Survivors (n = 549) | Non-survivors (n = 124) | p |
|---|---|---|---|---|
| Age (years) | 70.0 (60.0–79.0) | 67.0 (57.0–76.0) | 79.0 (72.0–87.0) | <0.0001 |
| Female | 312 (46.4) | 254 (37.7) | 58 (46.8) | 0.9184 |
| Length of hospitalization (days) | 13.0 (10.0–17.0) | 13.0 (10.0–17.0) | 12.0 (7.5–20.5) | 0.1773 |
| Hypertension | 413 (61.4) | 323 (58.8) | 90 (72.6) | 0.0045 |
| Coronary artery disease | 110 (16.3) | 82 (14.9) | 28 (22.6) | 0.0376 |
| Heart failure | 94 (14.0) | 60 (10.9) | 34 (27.4) | <0.0001 |
| Diabetes mellitus type 2 | 194 (28.8) | 150 (27.3) | 44 (35.5) | 0.0700 |
| Asthma / COPD | 72 (10.7) | 61 (11.1) | 11 (8.9) | 0.4661 |
| Obesity (BMI $\geq$30 kg/m$^2$) | 172 (25.6) | 143 (26.0) | 29 (23.4) | 0.5396 |
| Cancer | 31 (4.6) | 23 (4.2) | 8 (6.5) | 0.2778 |
| Oxygen saturation | 92.0 (88.0–95.0) | 93.0 (89.0–96.0) | 88.0 (80.0–93.0) | <0.0001 |
| Oxygen therapy | 598 (88.9) | 474 (86.3) | 124 (100.0) | 0.0001 |
| Non-invasive ventilation | 103 (15.3) | 55 (10.0) | 48 (38.7) | <0.0001 |
| Mechanical ventilation | 86 (12.8) | 20 (4.0) | 66 (53.2) | <0.0001 |
| Hemoglobin (g/dL) | 13.2 (11.7–14.3) | 13.3 (11.7–14.4) | 13.1 (11.4–14.2) | 0.3005 |
| Leukocytes (G/L) | 6.2 (4.5–8.8) | 6.1 (4.5–8.5) | 7.0 (4.9–10.4) | 0.0130 |
| Lymphocytes (%) | 1.4 (0.9–7.0) | 1.4 (0.9–7.1) | 1.3 (0.7–6.0) | 0.0539 |
| Neutrocytes (%) | 75.0 (64.6–83.5) | 73.0 (63.4–82.6) | 81.6 (73.0–88.8) | <0.0001 |
| Platelets (G/L) | 210.0 (158.0–280.5) | 220.5 (163.0–285.0) | 189.5 (132.5–262.0) | 0.0014 |
| C-reactive protein (mg/L) | 58.9 (22.6–120.8) | 53.2 (19.3–117.9) | 79.4 (43.9–138.6) | 0.0004 |
| Alanine amintransferase (U/L) | 30.5 (20.0–49.0) | 30.0 (20.0–49.0) | 31.0 (17.0–49.0) | 0.5056 |
| D-dimer (µg/L) | 973.5 (555.0–1878.0) | 910.0 (520.0–1659.0) | 1380.0 (720.0–2812.0) | 0.0001 |
| Glucose (mg/dL) | 119.0 (103.0–143.0) | 116.0 (100.0–137.0) | 134.0 (112.0–170.0) | <0.0001 |
| Creatinine (mg/dL) | 0.9 (0.7–1.2) | 0.9 (0.7–1.1) | 1.2 (0.8–1.6) | <0.0001 |
| eGFR | 78.8 (57.1–95.9) | 83.0 (63.0–99.3) | 58.5 (35.3–76.7) | <0.0001 |
| NT-proBNP (pg/mL) | 525.0 (166.0–1976.0) | 410.0 (137.0–1501.0) | 1927.5 (570.0–5903.0) | <0.0001 |
| Troponin T (ng/L) | 15.0 (7.1–39.0) | 12.7 (6.4–27.9) | 36.7 (16.0–80.0) | <0.0001 |
| Ferritin (µg/L) | 417.5 (217.5–851.0) | 388.0 (211.0–787.0) | 548.0 (269.0–1500.0) | 0.0395 |
| % of lung involvment in chest CT | 20.0 (10.0–40.0) | 20.0 (10.0–35.0) | 35.0 (12.5–60.0) | <0.0001 |

Abbreviations: BMI–body mass index; COPD–chronic obstructive pulmonary disease; CT–computed tomography; eGFR–estimated glomerular filtration rate; NT-pro BNP–N-terminal pro-brain natriuretic peptide.

solid organ or blood stem cell transplant, tuberculosis, and substance use disorder [24]. Early stratification of patients admitted due to COVID-19 and identification of those burdened with the highest risk of death can facilitate clinical decision-making without unnecessary postponement of intensive care treatment. Thus, we aimed to create a simple and user-friendly scoring

**Table 2. Results of the mulivariate logistic regression analysis.** Factors predictive of death at p<0.05.

| Parameter | OR | -95%CI | +95%CI | p |
|---|---|---|---|---|
| Age | 1.0.85 | 1.063 | 1.108 | <0.001 |
| SpO$_2$ | 0.947 | 0.920 | 0.975 | <0.001 |
| % of lung involvement in chest CT | 1.018 | 1.007 | 1.029 | 0.001 |
| Coronary artery disease | 1.047 | 0.611 | 1.795 | 0.867 |

Abbreviations: CI, confidence interval; CT, computed tomography; OR, odds ratio; SpO$_2$, oxygen saturation.

**Table 3. The 123 COVID SCORE.** Step 1 –provide patient's age; step 2 –measure oxygen saturation without oxygen supplementation; step 3 –assess the percentage of lung involvement in chest CT.

| Parameter | Points |
|---|---|
| **Age (years)** | |
| <63 | 0 |
| 63–82 | 6 |
| >82 | 9 |
| **SpO$_2$ (on admission, without oxygen supplementation)** | |
| >93 | 0 |
| 87–93 | 1 |
| <87 | 4 |
| **% of lung involvement in chest CT** | |
| <45 | 0 |
| 45–49 | 1 |
| 50–69 | 2 |
| >69 | 4 |

Abbreviations: CT, computed tomography; SpO$_2$, oxygen saturation.

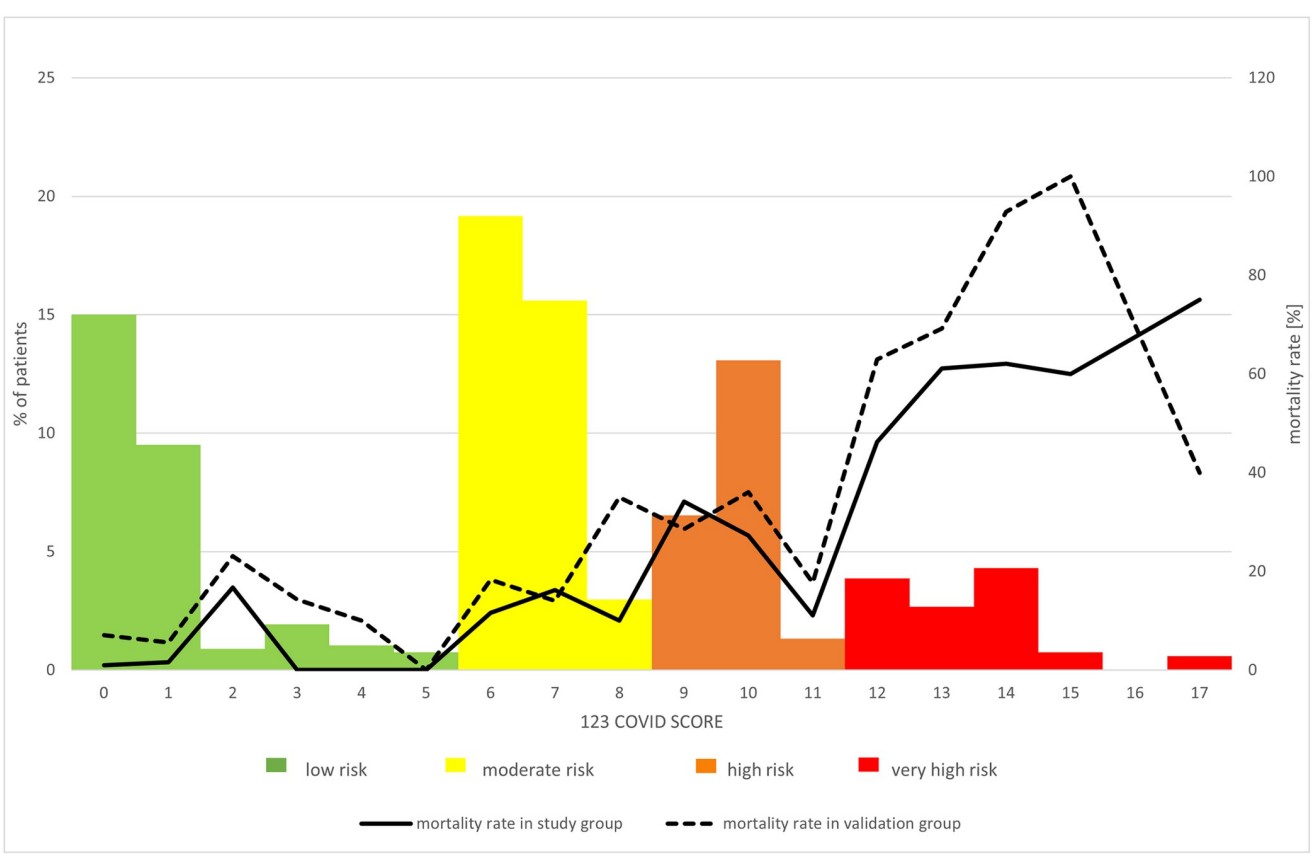

**Fig 1. The 123 COVID SCORE.** Histogram presenting percentage of patients from the study group receiving particular numbers of points and classification to low, moderate, high or very high risk of in-hospital death. Mortality rates are provided for study and validation groups.

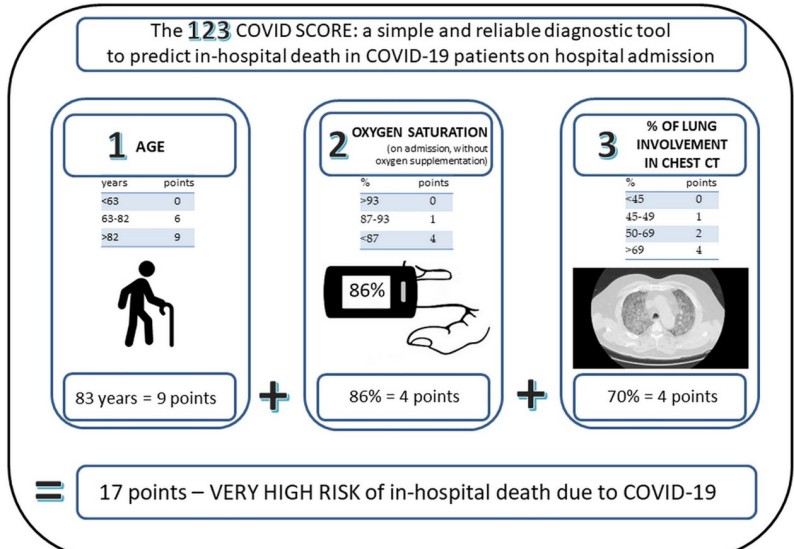

**Fig 2. How to calculate the 123 COVID SCORE?.**

system including a minimal number of parameters, but yet providing reliable and valid results. The 123 COVID SCORE includes age, $SpO_2$ on hospital admission measured without oxygen supplementation and percentage of lung involvement evaluated with chest CT. Created in a population of 673 COVID-19 patients and validated in a group of 558 patients, the 123 COVID SCORE offers a solid alternative to other scoring systems proposed so far.

The so far available scoring systems frequently use multiple laboratory tests and patient clinical characteristics to predict the risk of in-hospital death. However, a significant number of these variables are not routinely evaluated on hospital admission, which poses a major limitation for these scores.

One of the first clinical risk scores created to predict critical illness in patients hospitalized due to COVID-19 was the Chinese COVID-GRAM score [25]. Developed in a group of 1590 patients and validated in a cohort of 710 patients, it aimed to predict the need for intensive care and mechanical ventilation and assess the risk of death. Out of 72 variables included in the primary analysis, only 10 were incorporated in the risk score after being identified as predictors of critical illness in a logistic regression model. These variables encompass: chest X-ray abnormality, age, hemoptysis, dyspnea, unconsciousness, number of comorbidities, cancer history, neutrophil-to-lymphocyte ratio, lactate dehydrogenase concentration and direct bilirubin concentration. An on-line calculator was established to assess the risk of critical illness in COVID-19 patients. The accuracy of the COVID-GRAM score according to the AUROC was 0.88 (95% CI 0.85–0.91). Despite its high calculated accuracy, the COVID-GRAM score includes some variables that are imprecisely defined, for example chest X-ray abnormality, number of comorbidities or unconsciousness. It is important to notice that the COVID--GRAM score was created at the very beginnig of the COVID-19 pandemic in patients hospitalized from November 21, 2019 to January 31, 2020.

The Italian response was introduction of the National Early Warning Score 2 (NEWS2) by Gidari et al.–a score proposed for improved prediction of the critical course of COVID-19 in comparison to the COVID-GRAM score [26]. The NEWS2 score was developed at the very beginning of the COVID-19 pandemic in Europe in 71 patients hospitalized from March 1,

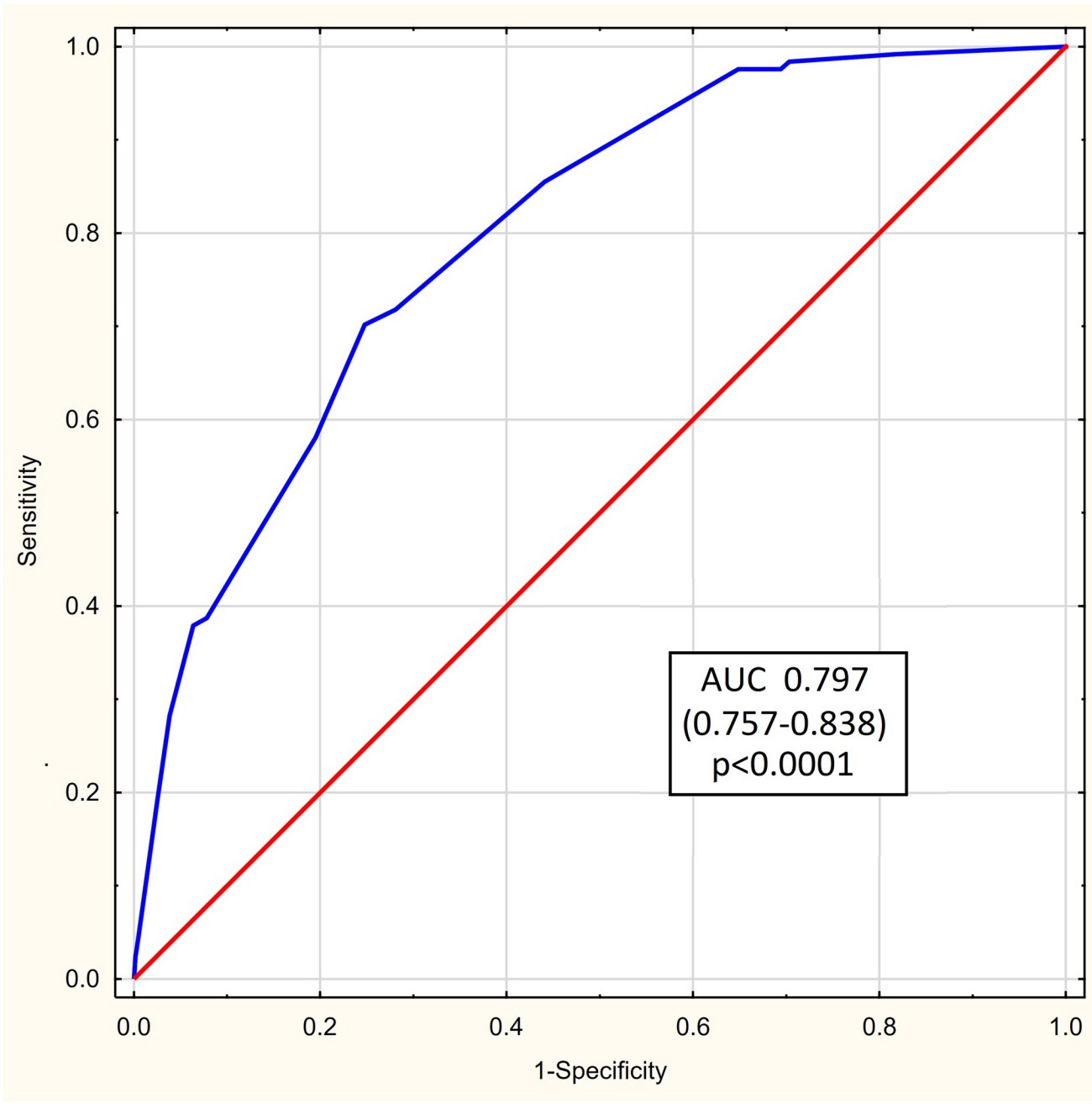

**Fig 3. The 123 COVID SCORE accuracy in the study group measured with the area under receiver operating curve (AUROC).**

2020 to April 20, 2020. The score included the following factors correlated with intensive care admission in multivariate analysis: respiratory rate, hypercapnic respiratory failure, oxygen supplementation, body temperature, systolic blood pressure, pulse rate and level of consciousness. The NEWS2 score accurately predicts intensive care admission with the ROC curve analysis AUC of 0.90 (95% CI 0.82–0.97). In a comparison of the COVID-GRAM vs. NEWS2 score performed in 121 Italian patients, the AUROC for NEWS2 was 0.87 (95% CI 0.80–0.93) and 0.77 (95% CI 0.68–0.85) for the COVID-GRAM score [27].

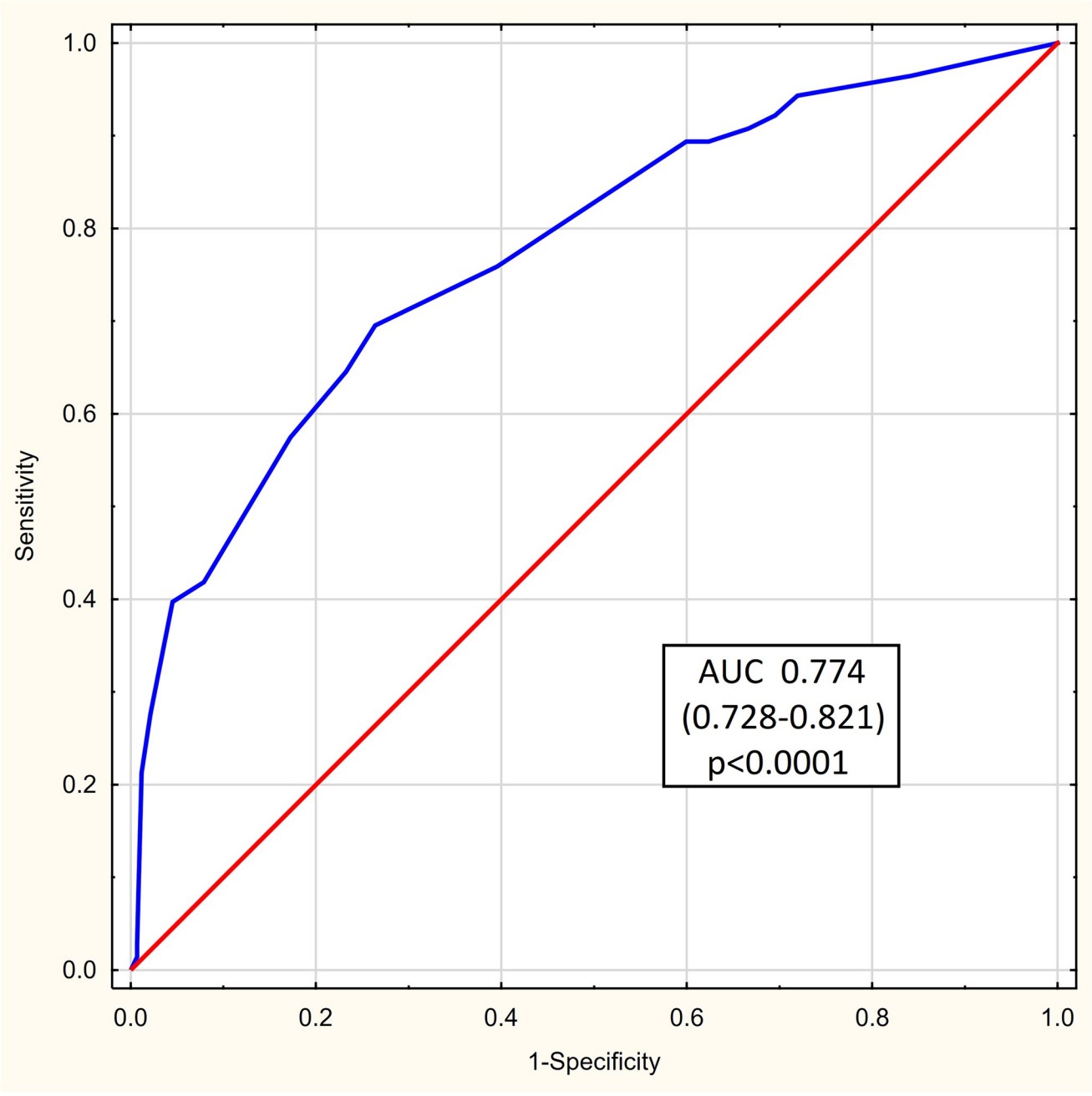

**Fig 4. The 123 COVID SCORE accuracy in the validation group measured with the area under receiver operating curve (AUROC).**

The main limitation of these two risk scores is their establishment at the very onset of the first wave of the COVID-19 pandemic, when the clinical course of the disease was largely dependent on the intensity of the cytokine storm.

The next in the gallery is the Covichem risk score composed of clinical and laboratory predictors of COVID-19 severity and developed by Bats et al. [28]. After a retrospective analysis of 303 COVID-19 patients with the use of a stepwise model selection by Akaike Information Criterion, 7 variables were identified in logistic regression model to predict prognosis of

COVID-19 patients. The Covichem score including: obesity, cardiovascular comorbidities, plasma sodium, albumin, ferritin, lactate dehydrogenase and creatine kinase has prediction accuracy of 0.87 (95% confidence interval 0.80–0.91). The score was validated in a group of 100 patients, exhibiting accuracy of 0.92 (95% CI 0.85–0.97); sensitivity of 0.89 and specificity of 0.95. It was impossible for us to calculate the Covichem risk score as albumin, ferritin and creatine kinase are not routinely assessed in our population.

The next on the list is the COVID-19 Lab score, developed to predict mortality using the following 11 laboratory tests obtained on hospital admission: haemoglobin, erythrocytes, leukocytes, neutrophils, lymphocytes, creatinine, C-reactive protein, interleukin-6, procalcitonin, lactate dehydrogenase, and D-dimer, selected according to the results of a multivariate logistic regression model [29]. The cut-off points for each variable were established based on ROC curves. The score assigned for each variable was based on odds ratios and ranged from 1 to 5. The total COVID-19 Lab score can range from 0 to 30 points. Low mortality risk was defined as a score of <12 points; moderate risk corresponded to 12 to 18 points and high risk to ≥19 points. In a group of 893 patients in whom the COVID-19 Lab score was created, in-hospital mortality rate was 3.9% in the low risk group, 16.1% in the moderate risk group and 49.1% in the high risk group. Due to missing data on interleukin-6 concentration on hospital admission, it was not possible to calculate the COVID-19 Lab score in our population.

Finally, a chest X-ray-based scoring system called the Brixia score proposed by Borghesi and Maroldi was developed to assess severity and progression of COVID-19 pneumonia [30]. After dividing each lung into 3 equal zones, a grade from 0 to 3 was assigned to each zone depending on the severity of lung abnormalities (0-no abnormalities; 1 –interstitial changes; 2 –interstitial and alveolar changes with interstitial predominance; 3-interstitial and alveolar changes with alveolar predominance), providing a total score of 0–18. The Brixia score was validated among 100 hospitalized patients with COVID-19. It renders significantly higher results in patients who died of COVID-19, compared with survivors (p≤0.02). In a subsequent study the authors demonstrated the Brixia score to correlate strongly with pneumonia severity and prognosis in 953 COVID-19 patients [31].

In a study by Aziz-Ahari et al. including 148 SARS-CoV2 positive Iranian patients a chest CT-based score was proposed to assess the severity of COVID-19, risk of intensive care unit admission, intubation requirement and death [32]. The chest CT severity score is calculated based on the degree of lung involvement by COVID-19, accounting for 0–4 points in each lung lobe depending on the percentage of lung involvement (0%–0 points; 1–25%–1 point; 26–50%–2 points; 51–75%–3 points; 76–100%–4 points) altogether resulting in 0–20 points. Based on the ROC curve analysis, an AUC of 0.726 was found to discriminate between recovery and death and the proposed CT score threshold was 15.5 with 61.8% sensitivity and 76.3% specificity. We found it impossible to calculate the CT severity score, because in our population of patients the percentage of lung involvement was evaluated collectively for each lung, rather than for each lobe separately.

The vast majority of the cited risk scores were established at the very beginning of the COVID-19 pandemic during the first pandemic wave. Thus, the generalizability of these risk scores to predict severe course of COVID-19 during the next waves of the pandemic and currently in post-pandemic time remains unknown. Moreover, development in different populations, healthcare systems and pandemic circumstances may also limit their usefulness. Additionally, incorporation of laboratory tests that are not routinely performed on hospital admission [28] or use of imprecise criteria [25] can either eliminate a risk score from practical applications or undermine the credibility of the results. Finally, physicians dealing with COVID-19 patients, many of the latter being critically ill, often have very limited time resources, not allowing to calculate complex, multi-variable risk scores.

## Limitations

There are a few limitations of this study to mention. Firstly, this is a retrospective analysis of prospectively collected data and due to missing data a substantial proportion of consecutive COVID-19 patients hospitalized in two Polish University Hospitals was removed from the analysis. Secondly, besides patients requiring hospitalization primarily due to COVID-19, the study group also comprised patients with concomitant COVID-19, hospitalized due to other reasons. Thirdly, in many regions of the world chest CT scans are not routinely performed in all COVID-19 patients requiring hospitalization, which may limit feasibility of our score. Fourthly, the 123 COVID SCORE was developed exclusively based on patients of Polish origin, from University Hospitals in two regions of Poland and validated in patients from 3 hospitals from 3 different regions, thus it requires additional testing in other ethnicities from different geographical regions.

## Conclusions

The 123 COVID SCORE including only 3 variables: age, SpO2 and percentage of lung involvement in chest CT, is a simple and reliable diagnostic tool to rapidly assess the risk of in-hospital death in COVID-19 patients upon hospital admission.

## Author Contributions

**Conceptualization:** Małgorzata Ostrowska, Aldona Kubica, Jacek Kubica.

**Data curation:** Małgorzata Ostrowska, Michał Kasprzak, Jacek Gajda, Natalia Jaje-Rykowska, Piotr Michalski, Beata Moczulska, Paulina Nowek, Maciej Piasecki, Marta Pilaczyńska-Cemel, Przemysław Podhajski, Paulina Prudzic, Dominika Stępniak, Damian Świątkowski, Maciej Żechowicz.

**Formal analysis:** Małgorzata Ostrowska, Michał Kasprzak, Tomasz Fabiszak.

**Investigation:** Małgorzata Ostrowska, Michał Kasprzak.

**Methodology:** Małgorzata Ostrowska, Michał Kasprzak.

**Project administration:** Małgorzata Ostrowska.

**Resources:** Jacek Kryś, Jacek Kubica.

**Software:** Michał Kasprzak.

**Supervision:** Robert Gajda, Leszek Gromadziński, Jacek Kryś, Aldona Kubica, Grzegorz Przybylski, Paweł Szymański, Jacek Kubica.

**Validation:** Małgorzata Ostrowska, Michał Kasprzak.

**Visualization:** Małgorzata Ostrowska, Michał Kasprzak.

**Writing – original draft:** Małgorzata Ostrowska.

**Writing – review & editing:** Michał Kasprzak, Tomasz Fabiszak, Jacek Gajda, Natalia Jaje-Rykowska, Piotr Michalski, Beata Moczulska, Paulina Nowek, Maciej Piasecki, Marta Pilaczyńska-Cemel, Przemysław Podhajski, Paulina Prudzic, Dominika Stępniak, Damian Świątkowski, Maciej Żechowicz, Robert Gajda, Leszek Gromadziński, Jacek Kryś, Aldona Kubica, Grzegorz Przybylski, Paweł Szymański, Jacek Kubica.

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
