## [Decision Letter · Decision Letter 0]

16 Jun 2024

PONE-D-24-02973The 123 COVID SCORE: a simple and reliable diagnostic tool to predict in-hospital death in COVID-19 patients on hospital admissionPLOS ONE

Dear Dr. Ostrowska,

Thank you for submitting your manuscript to PLOS ONE. After careful consideration, we feel that it has merit but does not fully meet PLOS ONE’s publication criteria as it currently stands. Therefore, we invite you to submit a revised version of the manuscript that addresses the points raised during the review process.

Please revise.

We look forward to receiving your revised manuscript.

Kind regards,

Academic Editor

PLOS ONE

Journal Requirements:

2. In this instance it seems there may be acceptable restrictions in place that prevent the public sharing of your minimal data. However, in line with our goal of ensuring long-term data availability to all interested researchers, PLOS’ Data Policy states that authors cannot be the sole named individuals responsible for ensuring data access (http://journals.plos.org/plosone/s/data-availability#loc-acceptable-data-sharing-methods).

Reviewers' comments:

Reviewer's Responses to Questions

**Comments to the Author**

1. Is the manuscript technically sound, and do the data support the conclusions?

Reviewer #1: Yes

Reviewer #2: Yes

2. Has the statistical analysis been performed appropriately and rigorously? 

Reviewer #1: I Don't Know

Reviewer #2: Yes

3. Have the authors made all data underlying the findings in their manuscript fully available?

Reviewer #1: Yes

Reviewer #2: Yes

4. Is the manuscript presented in an intelligible fashion and written in standard English?

Reviewer #1: Yes

Reviewer #2: Yes

5. Review Comments to the Author

Reviewer #1: I have reviewed the submitted article titled "Risk score predicting in-hospital mortality in COVID-19 patients on hospital admission" and find it to be well-written and highly informative. The study successfully develops a 123 COVID SCORE based on three key predictors - age, oxygen saturation, and percentage of lung involvement - to accurately assess the risk of in-hospital death among COVID-19 patients upon admission.

However, in order to further strengthen the clinical relevance and validity of the findings, I recommend seeking input from a specialist in infectious diseases for the clinical infectious disease section of the article, as well as from a statistical expert for the statistical modeling section. Incorporating their feedback and expertise would enhance the robustness of the study and provide additional insights for readers.

Overall, I believe this article makes a significant contribution to the field and with the suggested revisions, it has the potential to be even more impactful.

Minor comments:

I would like to mention that it is necessary to include a brief explanation alongside the titles of the tables. Additionally, I need to point out several changes in the writing style that should be considered for improving clarity and readability.

Reviewer #2: The coronavirus epidemic was a pivotal event in history, and patients are still burdened with a high risk of death. This manuscript aims to create a risk score consisting of three variables (age, oxygen saturation and percentage of lung involvement assessed with chest CT) predicting hospital mortality in patients on hospital admission.

6. PLOS authors have the option to publish the peer review history of their article (what does this mean?). If published, this will include your full peer review and any attached files.

Reviewer #1: No

Reviewer #2: No

---

## [Author Response · Author response to Decision Letter 0]

14 Jul 2024

11/07/2024

Academic Editor

PLOS ONE

Dear Editor,

 We greatly appreciate the careful review of our manuscript. We have done our best to resolve the issues raised by the Reviewers, and have made appropriate changes accordingly. 

Please find attached the detailed reply to all Reviewers’ comments and suggestions. Changes made in the revised manuscript file are tracked using the ‘Track changes’ mode as requested. We also confirm that our article meets all the Journal requirements. We provided two non-author institutional point of contact, as requested.

 We trust that you will find these changes satisfactory and we look forward to hearing from you at your earliest convenience.

Sincerely,

Corresponding author 

Małgorzata Ostrowska, MD, PhD

Department of Cardiology and Internal Medicine, Collegium Medicum, Nicolaus Copernicus University, 9 Skłodowskiej-Curie Street, 85-094 Bydgoszcz, Poland; 

E-mail: m.ostrowska@cm.umk.pl; Tel.: +48 52 5854023, Fax: +48 52 5854024;

Reviewer #1. 

1. I have reviewed the submitted article titled "Risk score predicting in-hospital mortality in COVID-19 patients on hospital admission" and find it to be well-written and highly informative. The study successfully develops a 123 COVID SCORE based on three key predictors - age, oxygen saturation, and percentage of lung involvement - to accurately assess the risk of in-hospital death among COVID-19 patients upon admission.

Answer: We would like to thank the Reviewer for favorable evaluation of our work. 

2. However, in order to further strengthen the clinical relevance and validity of the findings, I recommend seeking input from a specialist in infectious diseases for the clinical infectious disease section of the article, as well as from a statistical expert for the statistical modeling section.

Answer: We thank the Reviewer for this comment. It is extremely important for us to strengthen the clinical relevance and validity of our findings. That is why we went through all the current knowledge on COVID-19 from the World Health Organization, European Society of Clinical Microbiology and infectious Diseases Guidelines, European Centre for Disease Prevention and Control, as well as the United States Centers for Disease Prevention and Control to provide input from a world class specialists in infectious diseases. Although neither European, nor American guidelines recommend use of any particular score to assess the risk of severe COVID-19, they distinguish certain group of patients burdened with higher risk of severe disease. We enriched the Introduction and Discussion sections of the article. We added to Introduction: ‘’ According to WHO severity definitions we distinguish: i) critical COVID-19 defined by the criteria for acute respiratory distress syndrome, sepsis, septic shock or any condition requiring life sustaining treatment; ii) severe COVID-19 defined by oxygen saturation <90% on room air, signs of pneumonia or signs of severe respiratory distress; iii) non-severe COVID-19 defined as absence of any criteria for severe or critical COVID-19 [9]. (…) According to the European Centre for Disease Prevention and Control the risk of very severe disease increases with advancing age, presence of comorbidities and male sex [17].” (as marked in the track changes mode.) We added to Discussion: ‘’ Besides advanced age and male sex, the European Centre for Disease Prevention and Control lists hypertension, diabetes, chronic kidney disease, coronary heart disease, chronic obstructive pulmonary disease, cerebrovascular disease and chronic liver disease, use of immunosuppressive medications, arrythmia, ischemic heart disease, heart failure, cancer, obesity, and smoking as risk factors for severe disease [17]. The United States Centers for Disease Control and Prevention add few other medical conditions increasing the risk of severe COVID-19 as follows: cystic fibrosis, dementia or other neurological conditions, pregnancy, disability, human immunodeficiency virus infection, mental health conditions, physical inactivity, sickle cell disease or thalassemia, solid organ or blood stem cell transplant, tuberculosis, and substance use disorder [24].’’ (as marked in the track changes mode).

According to second part of the Reviewer’s comment to seek input from a statistical expert for the statistical modeling section we would like to mention Prof. Ewout W. Steyerberg from Leiden University Medical Center, Netherlands, author of a book on Clinical Prediction Models, leader in the area of Clinical Biostatistics and Clinical Decision Making. According to Prof. Ewout W. Steyeberg modern statistics using regression methods is the best way to create clinical prediction models. Practical tools providing innovative stepwise approach aid in development of accurate prediction models. We used the multivariable logistic regression model in order to identify predictors for mortality. In the next step a scoring module proposed by the Statistica package allowed us to use a structurized stepwise approach to create a scale predicitve of death eliminating the risk of making a mistake in calculations using old-fashioned data hungry methods. Our clinical prediction model, the 123 COVID SCORE was also externally validated in a population of 558 COVID-19 patients confirming its validity and reliability. The person responsible for creating our clincial prediction model is the second co-author, Michał Kasprzak, MD, PhD, who is highly experienced and certified in medical statistics, also in the creation of prediction models in science. To further explain the description of statistical methods used and made them fully reproducible we extended the statistical analysis section. We focused on creation a prediction model. Now it reads as follows: ’’To create a score for prediction of mortality risk, a scoring module of Statistica package was used. To select variables for building the model, the backward stepwise logistic regression method was used. Then, continuous variables were reclassified using the weight of evidence method. A scoring table was built, based on which the final prediction model was created.”

We hope that our efforts answer the concerns raised by the Reviewer. In case the Reviewer requires any further modifications please specify in detail.

Minor comments:

1. I would like to mention that it is necessary to include a brief explanation alongside the titles of the tables.

Answer: We would like to thank the Reviewer for this comment. We provided a brief explanation alongside the titles of the tables as recommended.

The title of the table 1 reads as follows: Table 1. Baseline clinical characteristics of study group. Variables are presented as median (interquartile range) or number (%).

The title of the Table 2 now reads as follows: Results of the mulivariate logistic regression analysis. Factors predictive of death at p<0.05. 

The title of the Table 3 now reads as follows: The 123 COVID SCORE. 1st step – provide patients’ age; 2nd step – measure oxygen saturation without oxygen supplementation; 3rd step – assess the percentage of lung involvement in chest CT. 

2. Additionally, I need to point out several changes in the writing style that should be considered for improving clarity and readability.

Answer: We did our best to improve our writing style and introduced many stylistic and grammatical improvements to enhance the clarity and readability of the article as recommended by the Reviewer (all improvements are marked in the track changes file).

Reviewer #2.

The coronavirus epidemic was a pivotal event in history, and patients are still burdened with a high risk of death. This manuscript aims to create a risk score consisting of three variables (age, oxygen saturation and percentage of lung involvement assessed with chest CT) predicting hospital mortality in patients on hospital admission.

Answer: We thank the Reviewer for this comment.

Journal Requirements:

Ad 1. We confirm that our article meets PLOS ONE's style requirements.

Ad 2. We provide two non-author institutional point of contact: 

1. kikkardiol@cm.umk.pl / tel. 0048525854023

This is an e-mail address and telephone number to Department of Cardiology and Internal Medicine, Collegium Medicum, Nicolaus Copernicus University, Bydgoszcz, Poland.

2. piotr.adamski@cm.umk.pl / tel. 0048525854023

This is an e-mail address and telephone number to non-author Associate professor in Department of Cardiology and Internal Medicine, Collegium Medicum, Nicolaus Copernicus University, Bydgoszcz, Poland.

We will ensure persistent and long-term data storage and availability with full data set deposited and available by contact with either corresponding author or two provided non-author contacts. One of them being institutional and durable point of contact, that ensures data will be accessible even if an author changes email addresses, institutions, or becomes unavailable to answer requests.

Ad 3. We confirm that the reference list is complete and correct.

---

## [Decision Letter · Decision Letter 1]

21 Aug 2024

The 123 COVID SCORE: a simple and reliable diagnostic tool to predict in-hospital death in COVID-19 patients on hospital admission

PONE-D-24-02973R1

Dear Dr. Ostrowska,

We’re pleased to inform you that your manuscript has been judged scientifically suitable for publication and will be formally accepted for publication once it meets all outstanding technical requirements.

Kind regards,

Academic Editor

PLOS ONE

Additional Editor Comments (optional):

Reviewers' comments:

Reviewer's Responses to Questions

**Comments to the Author**

1. If the authors have adequately addressed your comments raised in a previous round of review and you feel that this manuscript is now acceptable for publication, you may indicate that here to bypass the “Comments to the Author” section, enter your conflict of interest statement in the “Confidential to Editor” section, and submit your "Accept" recommendation.

Reviewer #1: All comments have been addressed

Reviewer #2: All comments have been addressed

2. Is the manuscript technically sound, and do the data support the conclusions?

Reviewer #1: Yes

Reviewer #2: Yes

3. Has the statistical analysis been performed appropriately and rigorously? 

Reviewer #1: Yes

Reviewer #2: Yes

4. Have the authors made all data underlying the findings in their manuscript fully available?

Reviewer #1: Yes

Reviewer #2: Yes

5. Is the manuscript presented in an intelligible fashion and written in standard English?

Reviewer #1: Yes

Reviewer #2: Yes

6. Review Comments to the Author

Reviewer #1: Dear Authors,

Thank you for your thorough revisions and for addressing the feedback provided. I appreciate the incorporation of insights from infectious disease specialists and the statistical expert, which significantly enhances the clinical relevance and robustness of the study. The adjustments in the introduction and discussion sections, along with the refined statistical analysis, provide important context and clarity to your findings. The improvements to the writing style have also greatly increased the article's readability.

I am pleased to recommend your article titled "Risk score predicting in-hospital mortality in COVID-19 patients on hospital admission" for acceptance. Your work contributes valuable insights to the understanding of mortality risk in COVID-19 patients.

Thank you once again for your diligent efforts.

Reviewer #2: (No Response)

7. PLOS authors have the option to publish the peer review history of their article (what does this mean?). If published, this will include your full peer review and any attached files.

Reviewer #1: No

Reviewer #2: No

---

## [Editor Report · Acceptance letter]

27 Aug 2024

PONE-D-24-02973R1 

PLOS ONE

Dear Dr. Ostrowska, 

I'm pleased to inform you that your manuscript has been deemed suitable for publication in PLOS ONE. Congratulations! Your manuscript is now being handed over to our production team.

Kind regards, 

on behalf of

Dr. Robert Jeenchen Chen 

Academic Editor

PLOS ONE